# LightGBM: A Highly Efficient Gradient Boosting Decision Tree

**Guolin Ke**[1], **Qi Meng**[2], **Thomas Finley**[3], **Taifeng Wang**[1],
**Wei Chen**[1], **Weidong Ma**[1], **Qiwei Ye**[1], **Tie-Yan Liu**[1]
[1]Microsoft Research   [2]Peking University   [3] Microsoft Redmond
[1]{guolin.ke, taifengw, wche, weima, qiwye, tie-yan.liu}@microsoft.com;
[2]qimeng13@pku.edu.cn;   [3]tfinely@microsoft.com;

## Abstract

Gradient Boosting Decision Tree (GBDT) is a popular machine learning algorithm, and has quite a few effective implementations such as XGBoost and pGBRT. Although many engineering optimizations have been adopted in these implementations, the efficiency and scalability are still unsatisfactory when the feature dimension is high and data size is large. A major reason is that for each feature, they need to scan all the data instances to estimate the information gain of all possible split points, which is very time consuming. To tackle this problem, we propose two novel techniques: *Gradient-based One-Side Sampling* (GOSS) and *Exclusive Feature Bundling* (EFB). With GOSS, we exclude a significant proportion of data instances with small gradients, and only use the rest to estimate the information gain. We prove that, since the data instances with larger gradients play a more important role in the computation of information gain, GOSS can obtain quite accurate estimation of the information gain with a much smaller data size. With EFB, we bundle mutually exclusive features (i.e., they rarely take nonzero values simultaneously), to reduce the number of features. We prove that finding the optimal bundling of exclusive features is NP-hard, but a greedy algorithm can achieve quite good approximation ratio (and thus can effectively reduce the number of features without hurting the accuracy of split point determination by much). We call our new GBDT implementation with GOSS and EFB *LightGBM*. Our experiments on multiple public datasets show that, LightGBM speeds up the training process of conventional GBDT by up to over 20 times while achieving almost the same accuracy.

## 1   Introduction

Gradient boosting decision tree (GBDT) [1] is a widely-used machine learning algorithm, due to its efficiency, accuracy, and interpretability. GBDT achieves state-of-the-art performances in many machine learning tasks, such as multi-class classification [2], click prediction [3], and learning to rank [4]. In recent years, with the emergence of big data (in terms of both the number of features and the number of instances), GBDT is facing new challenges, especially in the tradeoff between accuracy and efficiency. Conventional implementations of GBDT need to, for every feature, scan all the data instances to estimate the information gain of all the possible split points. Therefore, their computational complexities will be proportional to both the number of features and the number of instances. This makes these implementations very time consuming when handling big data.

To tackle this challenge, a straightforward idea is to reduce the number of data instances and the number of features. However, this turns out to be highly non-trivial. For example, it is unclear how to perform data sampling for GBDT. While there are some works that sample data according to their weights to speed up the training process of boosting [5, 6, 7], they cannot be directly applied to GBDT

since there is no sample weight in GBDT at all. In this paper, we propose two novel techniques towards this goal, as elaborated below.

*Gradient-based One-Side Sampling* (GOSS). While there is no native weight for data instance in GBDT, we notice that data instances with different gradients play different roles in the computation of information gain. In particular, according to the definition of information gain, those instances with larger gradients[1] (i.e., under-trained instances) will contribute more to the information gain. Therefore, when down sampling the data instances, in order to retain the accuracy of information gain estimation, we should better keep those instances with large gradients (e.g., larger than a pre-defined threshold, or among the top percentiles), and only randomly drop those instances with small gradients. We prove that such a treatment can lead to a more accurate gain estimation than uniformly random sampling, with the same target sampling rate, especially when the value of information gain has a large range.

*Exclusive Feature Bundling* (EFB). Usually in real applications, although there are a large number of features, the feature space is quite sparse, which provides us a possibility of designing a nearly lossless approach to reduce the number of effective features. Specifically, in a sparse feature space, many features are (almost) exclusive, i.e., they rarely take nonzero values simultaneously. Examples include the one-hot features (e.g., one-hot word representation in text mining). We can safely bundle such exclusive features. To this end, we design an efficient algorithm by reducing the optimal bundling problem to a graph coloring problem (by taking features as vertices and adding edges for every two features if they are not mutually exclusive), and solving it by a greedy algorithm with a constant approximation ratio.

We call the new GBDT algorithm with GOSS and EFB *LightGBM*[2]. Our experiments on multiple public datasets show that LightGBM can accelerate the training process by up to over 20 times while achieving almost the same accuracy.

The remaining of this paper is organized as follows. At first, we review GBDT algorithms and related work in Sec. 2. Then, we introduce the details of GOSS in Sec. 3 and EFB in Sec. 4. Our experiments for LightGBM on public datasets are presented in Sec. 5. Finally, we conclude the paper in Sec. 6.

## 2 Preliminaries

### 2.1 GBDT and Its Complexity Analysis

GBDT is an ensemble model of decision trees, which are trained in sequence [1]. In each iteration, GBDT learns the decision trees by fitting the negative gradients (also known as residual errors).

The main cost in GBDT lies in learning the decision trees, and the most time-consuming part in learning a decision tree is to find the best split points. One of the most popular algorithms to find split points is the pre-sorted algorithm [8, 9], which enumerates all possible split points on the pre-sorted feature values. This algorithm is simple and can find the optimal split points, however, it is inefficient in both training speed and memory consumption. Another popular algorithm is the histogram-based algorithm [10, 11, 12], as shown in Alg. 1[3]. Instead of finding the split points on the sorted feature values, histogram-based algorithm buckets continuous feature values into discrete bins and uses these bins to construct feature histograms during training. Since the histogram-based algorithm is more efficient in both memory consumption and training speed, we will develop our work on its basis.

As shown in Alg. 1, the histogram-based algorithm finds the best split points based on the feature histograms. It costs $O(\#data \times \#feature)$ for histogram building and $O(\#bin \times \#feature)$ for split point finding. Since #bin is usually much smaller than #data, histogram building will dominate the computational complexity. If we can reduce #data or #feature, we will be able to substantially speed up the training of GBDT.

### 2.2 Related Work

There have been quite a few implementations of GBDT in the literature, including XGBoost [13], pGBRT [14], scikit-learn [15], and gbm in R [16] [4]. Scikit-learn and gbm in R implements the pre-sorted algorithm, and pGBRT implements the histogram-based algorithm. XGBoost supports both

the pre-sorted algorithm and histogram-based algorithm. As shown in [13], XGBoost outperforms the other tools. So, we use XGBoost as our baseline in the experiment section.

To reduce the size of the training data, a common approach is to down sample the data instances. For example, in [5], data instances are filtered if their weights are smaller than a fixed threshold. SGB [20] uses a random subset to train the weak learners in every iteration. In [6], the sampling ratio are dynamically adjusted in the training progress. However, all these works except SGB [20] are based on AdaBoost [21], and cannot be directly applied to GBDT since there are no native weights for data instances in GBDT. Though SGB can be applied to GBDT, it usually hurts accuracy and thus it is not a desirable choice.

Similarly, to reduce the number of features, it is natural to filter weak features [22, 23, 7, 24]. This is usually done by principle component analysis or projection pursuit. However, these approaches highly rely on the assumption that features contain significant redundancy, which might not always be true in practice (features are usually designed with their unique contributions and removing any of them may affect the training accuracy to some degree).

The large-scale datasets used in real applications are usually quite sparse. GBDT with the pre-sorted algorithm can reduce the training cost by ignoring the features with zero values [13]. However, GBDT with the histogram-based algorithm does not have efficient sparse optimization solutions. The reason is that the histogram-based algorithm needs to retrieve feature bin values (refer to Alg. 1) for each data instance no matter the feature value is zero or not. It is highly preferred that GBDT with the histogram-based algorithm can effectively leverage such sparse property.

To address the limitations of previous works, we propose two new novel techniques called Gradient-based One-Side Sampling (GOSS) and Exclusive Feature Bundling (EFB). More details will be introduced in the next sections.

---

**Algorithm 1:** Histogram-based Algorithm

**Input**: $I$: training data, $d$: max depth
**Input**: $m$: feature dimension
$nodeSet \leftarrow \{0\}$ ▷ tree nodes in current level
$rowSet \leftarrow \{\{0, 1, 2, ...\}\}$ ▷ data indices in tree nodes
**for** $i = 1$ **to** $d$ **do**
    **for** $node$ **in** $nodeSet$ **do**
        usedRows $\leftarrow rowSet[node]$
        **for** $k = 1$ **to** $m$ **do**
            $H \leftarrow$ new Histogram()
            ▷ Build histogram
            **for** $j$ **in** $usedRows$ **do**
                bin $\leftarrow I$.f[k][j].bin
                $H$[bin].y $\leftarrow H$[bin].y + I.y[j]
                $H$[bin].n $\leftarrow H$[bin].n + 1
            Find the best split on histogram $H$.
        ...
    Update $rowSet$ and $nodeSet$ according to the best split points.
    ...

---

**Algorithm 2:** Gradient-based One-Side Sampling

**Input**: $I$: training data, $d$: iterations
**Input**: $a$: sampling ratio of large gradient data
**Input**: $b$: sampling ratio of small gradient data
**Input**: $loss$: loss function, $L$: weak learner
models $\leftarrow \{\}$, fact $\leftarrow \frac{1-a}{b}$
topN $\leftarrow$ a $\times$ len($I$) , randN $\leftarrow$ b $\times$ len($I$)
**for** $i = 1$ **to** $d$ **do**
    preds $\leftarrow$ models.predict($I$)
    g $\leftarrow loss(I,$ preds), w $\leftarrow \{1,1,...\}$
    sorted $\leftarrow$ GetSortedIndices(abs(g))
    topSet $\leftarrow$ sorted[1:topN]
    randSet $\leftarrow$ RandomPick(sorted[topN:len(I)], randN)
    usedSet $\leftarrow$ topSet + randSet
    w[randSet] $\times =$ fact ▷ Assign weight $fact$ to the small gradient data.
    newModel $\leftarrow$ L($I$[usedSet], $-$ g[usedSet], w[usedSet])
    models.append(newModel)

---

## 3 Gradient-based One-Side Sampling

In this section, we propose a novel sampling method for GBDT that can achieve a good balance between reducing the number of data instances and keeping the accuracy for learned decision trees.

### 3.1 Algorithm Description

In AdaBoost, the sample weight serves as a good indicator for the importance of data instances. However, in GBDT, there are no native sample weights, and thus the sampling methods proposed for AdaBoost cannot be directly applied. Fortunately, we notice that the gradient for each data instance in GBDT provides us with useful information for data sampling. That is, if an instance is associated with a small gradient, the training error for this instance is small and it is already well-trained. A straightforward idea is to discard those data instances with small gradients. However, the data distribution will be changed by doing so, which will hurt the accuracy of the learned model. To avoid this problem, we propose a new method called Gradient-based One-Side Sampling (GOSS).

GOSS keeps all the instances with large gradients and performs random sampling on the instances with small gradients. In order to compensate the influence to the data distribution, when computing the information gain, GOSS introduces a constant multiplier for the data instances with small gradients (see Alg. 2). Specifically, GOSS firstly sorts the data instances according to the absolute value of their gradients and selects the top $a \times 100\%$ instances. Then it randomly samples $b \times 100\%$ instances from the rest of the data. After that, GOSS amplifies the sampled data with small gradients by a constant $\frac{1-a}{b}$ when calculating the information gain. By doing so, we put more focus on the under-trained instances without changing the original data distribution by much.

### 3.2 Theoretical Analysis

GBDT uses decision trees to learn a function from the input space $\mathcal{X}^s$ to the gradient space $\mathcal{G}$ [1]. Suppose that we have a training set with $n$ i.i.d. instances $\{x_1, \cdots, x_n\}$, where each $x_i$ is a vector with dimension $s$ in space $\mathcal{X}^s$. In each iteration of gradient boosting, the negative gradients of the loss function with respect to the output of the model are denoted as $\{g_1, \cdots, g_n\}$. The decision tree model splits each node at the most informative feature (with the largest information gain). For GBDT, the information gain is usually measured by the variance after splitting, which is defined as below.

**Definition 3.1** *Let $O$ be the training dataset on a fixed node of the decision tree. The variance gain of splitting feature $j$ at point $d$ for this node is defined as*

$$V_{j|O}(d) = \frac{1}{n_O} \left( \frac{\left(\sum_{\{x_i \in O: x_{ij} \leq d\}} g_i\right)^2}{n_{l|O}^j(d)} + \frac{\left(\sum_{\{x_i \in O: x_{ij} > d\}} g_i\right)^2}{n_{r|O}^j(d)} \right),$$

*where $n_O = \sum I[x_i \in O]$, $n_{l|O}^j(d) = \sum I[x_i \in O : x_{ij} \leq d]$ and $n_{r|O}^j(d) = \sum I[x_i \in O : x_{ij} > d]$.*

For feature $j$, the decision tree algorithm selects $d_j^* = argmax_d V_j(d)$ and calculates the largest gain $V_j(d_j^*)$. [5] Then, the data are split according feature $j^*$ at point $d_{j*}$ into the left and right child nodes.

In our proposed GOSS method, first, we rank the training instances according to their absolute values of their gradients in the descending order; second, we keep the top-$a \times 100\%$ instances with the larger gradients and get an instance subset $A$; then, for the remaining set $A^c$ consisting $(1-a) \times 100\%$ instances with smaller gradients, we further randomly sample a subset $B$ with size $b \times |A^c|$; finally, we split the instances according to the estimated variance gain $\tilde{V}_j(d)$ over the subset $A \cup B$, i.e.,

$$\tilde{V}_j(d) = \frac{1}{n} \left( \frac{\left(\sum_{x_i \in A_l} g_i + \frac{1-a}{b} \sum_{x_i \in B_l} g_i\right)^2}{n_l^j(d)} + \frac{\left(\sum_{x_i \in A_r} g_i + \frac{1-a}{b} \sum_{x_i \in B_r} g_i\right)^2}{n_r^j(d)} \right), \tag{1}$$

where $A_l = \{x_i \in A : x_{ij} \leq d\}$, $A_r = \{x_i \in A : x_{ij} > d\}$, $B_l = \{x_i \in B : x_{ij} \leq d\}$, $B_r = \{x_i \in B : x_{ij} > d\}$, and the coefficient $\frac{1-a}{b}$ is used to normalize the sum of the gradients over $B$ back to the size of $A^c$.

Thus, in GOSS, we use the estimated $\tilde{V}_j(d)$ over a smaller instance subset, instead of the accurate $V_j(d)$ over all the instances to determine the split point, and the computation cost can be largely reduced. More importantly, the following theorem indicates that GOSS will not lose much training accuracy and will outperform random sampling. Due to space restrictions, we leave the proof of the theorem to the supplementary materials.

**Theorem 3.2** *We denote the approximation error in GOSS as $\mathcal{E}(d) = |\tilde{V}_j(d) - V_j(d)|$ and $\bar{g}_l^j(d) = \frac{\sum_{x_i \in (A \cup A^c)_l} |g_i|}{n_l^j(d)}, \bar{g}_r^j(d) = \frac{\sum_{x_i \in (A \cup A^c)_r} |g_i|}{n_r^j(d)}$. With probability at least $1 - \delta$, we have*

$$\mathcal{E}(d) \leq C_{a,b}^2 \ln 1/\delta \cdot \max \left\{ \frac{1}{n_l^j(d)}, \frac{1}{n_r^j(d)} \right\} + 2DC_{a,b} \sqrt{\frac{\ln 1/\delta}{n}}, \tag{2}$$

*where $C_{a,b} = \frac{1-a}{\sqrt{b}} \max_{x_i \in A^c} |g_i|$, and $D = \max(\bar{g}_l^j(d), \bar{g}_r^j(d))$.*

According to the theorem, we have the following discussions: (1) The asymptotic approximation ratio of GOSS is $\mathcal{O}\left( \frac{1}{n_l^j(d)} + \frac{1}{n_r^j(d)} + \frac{1}{\sqrt{n}} \right)$. If the split is not too unbalanced (i.e., $n_l^j(d) \geq \mathcal{O}(\sqrt{n})$ and $n_r^j(d) \geq \mathcal{O}(\sqrt{n})$), the approximation error will be dominated by the second term of Ineq.(2) which

decreases to 0 in $\mathcal{O}(\sqrt{n})$ with $n \to \infty$. That means when number of data is large, the approximation is quite accurate. (2) Random sampling is a special case of GOSS with $a = 0$. In many cases, GOSS could outperform random sampling, under the condition $C_{0,\beta} > C_{a,\beta-a}$, which is equivalent to $\frac{\alpha_a}{\sqrt{\beta}} > \frac{1-a}{\sqrt{\beta-a}}$ with $\alpha_a = \max_{x_i \in A \cup A^c} |g_i| / \max_{x_i \in A^c} |g_i|$.

Next, we analyze the generalization performance in GOSS. We consider the generalization error in GOSS $\mathcal{E}_{gen}^{GOSS}(d) = |\tilde{V}_j(d) - V_*(d)|$, which is the gap between the variance gain calculated by the sampled training instances in GOSS and the true variance gain for the underlying distribution. We have $\mathcal{E}_{gen}^{GOSS}(d) \leq |\tilde{V}_j(d) - V_j(d)| + |V_j(d) - V_*(d)| \triangleq \mathcal{E}_{GOSS}(d) + \mathcal{E}_{gen}(d)$. Thus, the generalization error with GOSS will be close to that calculated by using the full data instances if the GOSS approximation is accurate. On the other hand, sampling will increase the diversity of the base learners, which potentially help to improve the generalization performance [24].

# 4 Exclusive Feature Bundling

In this section, we propose a novel method to effectively reduce the number of features.

---

**Algorithm 3:** Greedy Bundling

**Input**: $F$: features, $K$: max conflict count
Construct graph $G$
searchOrder $\leftarrow$ G.sortByDegree()
bundles $\leftarrow$ {}, bundlesConflict $\leftarrow$ {}
**for** $i$ **in** *searchOrder* **do**
    needNew $\leftarrow$ True
    **for** $j = 1$ **to** *len(bundles)* **do**
        cnt $\leftarrow$ ConflictCnt(bundles[j],$F$[i])
        **if** *cnt + bundlesConflict[i]* $\leq K$ **then**
            bundles[j].add($F$[i]), needNew $\leftarrow$ False
            break
    **if** *needNew* **then**
        Add $F[i]$ as a new bundle to *bundles*

**Output**: *bundles*

---

**Algorithm 4:** Merge Exclusive Features

**Input**: $numData$: number of data
**Input**: $F$: One bundle of exclusive features
binRanges $\leftarrow$ {0}, totalBin $\leftarrow$ 0
**for** $f$ *in* $F$ **do**
    totalBin += f.numBin
    binRanges.append(totalBin)
newBin $\leftarrow$ new Bin(numData)
**for** $i = 1$ **to** $numData$ **do**
    newBin[i] $\leftarrow$ 0
    **for** $j = 1$ **to** len(F) **do**
        **if** $F[j].bin[i] \neq 0$ **then**
            newBin[i] $\leftarrow$ F[j].bin[i] + binRanges[j]

**Output**: $newBin$, $binRanges$

---

High-dimensional data are usually very sparse. The sparsity of the feature space provides us a possibility of designing a nearly lossless approach to reduce the number of features. Specifically, in a sparse feature space, many features are mutually exclusive, i.e., they never take nonzero values simultaneously. We can safely bundle exclusive features into a single feature (which we call an *exclusive feature bundle*). By a carefully designed feature scanning algorithm, we can build the same feature histograms from the feature bundles as those from individual features. In this way, the complexity of histogram building changes from $O(\#data \times \#feature)$ to $O(\#data \times \#bundle)$, while $\#bundle << \#feature$. Then we can significantly speed up the training of GBDT without hurting the accuracy. In the following, we will show how to achieve this in detail.

There are two issues to be addressed. The first one is to determine which features should be bundled together. The second is how to construct the bundle.

**Theorem 4.1** *The problem of partitioning features into a smallest number of exclusive bundles is NP-hard.*

*Proof:* We will reduce the graph coloring problem [25] to our problem. Since graph coloring problem is NP-hard, we can then deduce our conclusion.

Given any instance $G = (V, E)$ of the graph coloring problem. We construct an instance of our problem as follows. Take each row of the incidence matrix of $G$ as a feature, and get an instance of our problem with $|V|$ features. It is easy to see that an exclusive bundle of features in our problem corresponds to a set of vertices with the same color, and vice versa. $\square$

For the first issue, we prove in Theorem 4.1 that it is NP-Hard to find the optimal bundling strategy, which indicates that it is impossible to find an exact solution within polynomial time. In order to find a good approximation algorithm, we first reduce the optimal bundling problem to the graph coloring problem by taking features as vertices and adding edges for every two features if they are not mutually exclusive, then we use a greedy algorithm which can produce reasonably good results

(with a constant approximation ratio) for graph coloring to produce the bundles. Furthermore, we notice that there are usually quite a few features, although not 100% mutually exclusive, also rarely take nonzero values simultaneously. If our algorithm can allow a small fraction of conflicts, we can get an even smaller number of feature bundles and further improve the computational efficiency. By simple calculation, random polluting a small fraction of feature values will affect the training accuracy by at most $\mathcal{O}([(1-\gamma)n]^{-2/3})$(See Proposition 2.1 in the supplementary materials), where $\gamma$ is the maximal conflict rate in each bundle. So, if we choose a relatively small $\gamma$, we will be able to achieve a good balance between accuracy and efficiency.

Based on the above discussions, we design an algorithm for exclusive feature bundling as shown in Alg. 3. First, we construct a graph with weighted edges, whose weights correspond to the total conflicts between features. Second, we sort the features by their degrees in the graph in the descending order. Finally, we check each feature in the ordered list, and either assign it to an existing bundle with a small conflict (controlled by $\gamma$), or create a new bundle. The time complexity of Alg. 3 is $O(\#feature^2)$ and it is processed only once before training. This complexity is acceptable when the number of features is not very large, but may still suffer if there are millions of features. To further improve the efficiency, we propose a more efficient ordering strategy without building the graph: ordering by the count of nonzero values, which is similar to ordering by degrees since more nonzero values usually leads to higher probability of conflicts. Since we only alter the ordering strategies in Alg. 3, the details of the new algorithm are omitted to avoid duplication.

For the second issues, we need a good way of merging the features in the same bundle in order to reduce the corresponding training complexity. The key is to ensure that the values of the original features can be identified from the feature bundles. Since the histogram-based algorithm stores discrete bins instead of continuous values of the features, we can construct a feature bundle by letting exclusive features reside in different bins. This can be done by adding offsets to the original values of the features. For example, suppose we have two features in a feature bundle. Originally, feature A takes value from $[0,10)$ and feature B takes value $[0,20)$. We then add an offset of 10 to the values of feature B so that the refined feature takes values from $[10,30)$. After that, it is safe to merge features A and B, and use a feature bundle with range $[0,30]$ to replace the original features A and B. The detailed algorithm is shown in Alg. 4.

EFB algorithm can bundle many exclusive features to the much fewer dense features, which can effectively avoid unnecessary computation for zero feature values. Actually, we can also optimize the basic histogram-based algorithm towards ignoring the zero feature values by using a table for each feature to record the data with nonzero values. By scanning the data in this table, the cost of histogram building for a feature will change from $O(\#data)$ to $O(\#non\_zero\_data)$. However, this method needs additional memory and computation cost to maintain these per-feature tables in the whole tree growth process. We implement this optimization in LightGBM as a basic function. Note, this optimization does not conflict with EFB since we can still use it when the bundles are sparse.

## 5 Experiments

In this section, we report the experimental results regarding our proposed LightGBM algorithm. We use five different datasets which are all publicly available. The details of these datasets are listed in Table 1. Among them, the Microsoft Learning to Rank (LETOR) [26] dataset contains 30K web search queries. The features used in this dataset are mostly dense numerical features. The Allstate Insurance Claim [27] and the Flight Delay [28] datasets both contain a lot of one-hot coding features. And the last two datasets are from KDD CUP 2010 and KDD CUP 2012. We directly use the features used by the winning solution from NTU [29, 30, 31], which contains both dense and sparse features, and these two datasets are very large. These datasets are large, include both sparse and dense features, and cover many real-world tasks. Thus, we can use them to test our algorithm thoroughly.

Our experimental environment is a Linux server with two E5-2670 v3 CPUs (in total 24 cores) and 256GB memories. All experiments run with multi-threading and the number of threads is fixed to 16.

### 5.1 Overall Comparison

We present the overall comparisons in this subsection. XGBoost [13] and LightGBM without GOSS and EFB (called lgb_baselline) are used as baselines. For XGBoost, we used two versions, xgb_exa (pre-sorted algorithm) and xgb_his (histogram-based algorithm). For xgb_his, lgb_baseline, and LightGBM, we used the leaf-wise tree growth strategy [32]. For xgb_exa, since it only supports layer-wise growth strategy, we tuned the parameters for xgb_exa to let it grow similar trees like other

Table 1: Datasets used in the experiments.

| Name | $\#data$ | $\#feature$ | Description | Task | Metric |
|---|---|---|---|---|---|
| Allstate | 12 M | 4228 | Sparse | Binary classification | AUC |
| Flight Delay | 10 M | 700 | Sparse | Binary classification | AUC |
| LETOR | 2M | 136 | Dense | Ranking | NDCG [4] |
| KDD10 | 19M | 29M | Sparse | Binary classification | AUC |
| KDD12 | 119M | 54M | Sparse | Binary classification | AUC |

Table 2: Overall training time cost comparison. LightGBM is lgb_baseline with GOSS and EFB. EFB_only is lgb_baseline with EFB. The values in the table are the average time cost (seconds) for training one iteration.

| | xgb_exa | xgb_his | lgb_baseline | EFB_only | **LightGBM** |
|---|---|---|---|---|---|
| Allstate | 10.85 | 2.63 | 6.07 | 0.71 | **0.28** |
| Flight Delay | 5.94 | 1.05 | 1.39 | 0.27 | **0.22** |
| LETOR | 5.55 | 0.63 | 0.49 | 0.46 | **0.31** |
| KDD10 | 108.27 | OOM | 39.85 | 6.33 | **2.85** |
| KDD12 | 191.99 | OOM | 168.26 | 20.23 | **12.67** |

Table 3: Overall accuracy comparison on test datasets. Use AUC for classification task and NDCG@10 for ranking task. SGB is lgb_baseline with Stochastic Gradient Boosting, and its sampling ratio is the same as LightGBM.

| | xgb_exa | xgb_his | lgb_baseline | SGB | **LightGBM** |
|---|---|---|---|---|---|
| Allstate | 0.6070 | 0.6089 | 0.6093 | $0.6064 \pm 7e\text{-}4$ | **$0.6093 \pm 9e\text{-}5$** |
| Flight Delay | 0.7601 | 0.7840 | 0.7847 | $0.7780 \pm 8e\text{-}4$ | **$0.7846 \pm 4e\text{-}5$** |
| LETOR | 0.4977 | 0.4982 | 0.5277 | $0.5239 \pm 6e\text{-}4$ | **$0.5275 \pm 5e\text{-}4$** |
| KDD10 | 0.7796 | OOM | 0.78735 | $0.7759 \pm 3e\text{-}4$ | **$0.78732 \pm 1e\text{-}4$** |
| KDD12 | 0.7029 | OOM | 0.7049 | $0.6989 \pm 8e\text{-}4$ | **$0.7051 \pm 5e\text{-}5$** |

methods. And we also tuned the parameters for all datasets towards a better balancing between speed and accuracy. We set $a = 0.05, b = 0.05$ for Allstate, KDD10 and KDD12, and set $a = 0.1, b = 0.1$ for Flight Delay and LETOR. We set $\gamma = 0$ in EFB. All algorithms are run for fixed iterations, and we get the accuracy results from the iteration with the best score.[6]

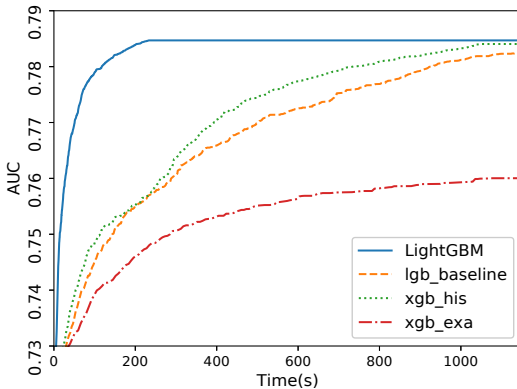

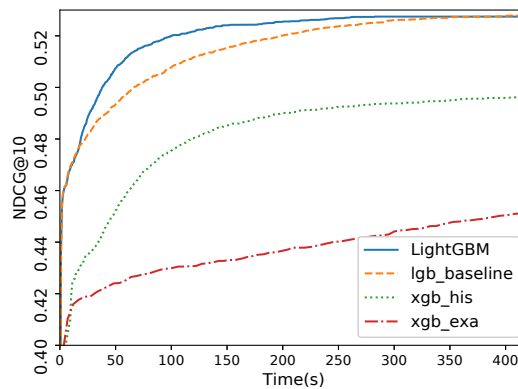

Figure 1: Time-AUC curve on Flight Delay.      Figure 2: Time-NDCG curve on LETOR.

The training time and test accuracy are summarized in Table 2 and Table 3 respectively. From these results, we can see that LightGBM is the fastest while maintaining almost the same accuracy as baselines. The xgb_exa is based on the pre-sorted algorithm, which is quite slow comparing with histogram-base algorithms. By comparing with lgb_baseline, LightGBM speed up 21x, 6x, 1.6x, 14x and 13x respectively on the Allstate, Flight Delay, LETOR, KDD10 and KDD12 datasets. Since xgb_his is quite memory consuming, it cannot run successfully on KDD10 and KDD12 datasets due to out-of-memory. On the remaining datasets, LightGBM are all faster, up to 9x speed-up is achieved on the Allstate dataset. The speed-up is calculated based on training time per iteration since all algorithms converge after similar number of iterations. To demonstrate the overall training process, we also show the training curves based on wall clock time on Flight Delay and LETOR in the Fig. 1

Table 4: Accuracy comparison on LETOR dataset for GOSS and SGB under different sampling ratios. We ensure all experiments reach the convergence points by using large iterations with early stopping. The standard deviations on different settings are small. The settings of $a$ and $b$ for GOSS can be found in the supplementary materials.

| Sampling ratio | 0.1 | 0.15 | 0.2 | 0.25 | 0.3 | 0.35 | 0.4 |
|---|---|---|---|---|---|---|---|
| SGB | 0.5182 | 0.5216 | 0.5239 | 0.5249 | 0.5252 | 0.5263 | 0.5267 |
| GOSS | 0.5224 | 0.5256 | 0.5275 | 0.5284 | 0.5289 | 0.5293 | 0.5296 |

and Fig. 2, respectively. To save space, we put the remaining training curves of the other datasets in the supplementary material.

On all datasets, LightGBM can achieve almost the same test accuracy as the baselines. This indicates that both GOSS and EFB will not hurt accuracy while bringing significant speed-up. It is consistent with our theoretical analysis in Sec. 3.2 and Sec. 4.

LightGBM achieves quite different speed-up ratios on these datasets. The overall speed-up comes from the combination of GOSS and EFB, we will break down the contribution and discuss the effectiveness of GOSS and EFB separately in the next sections.

### 5.2 Analysis on GOSS

First, we study the speed-up ability of GOSS. From the comparison of LightGBM and EFB_only (LightGBM without GOSS) in Table 2, we can see that GOSS can bring nearly 2x speed-up by its own with using 10% - 20% data. GOSS can learn trees by only using the sampled data. However, it retains some computations on the full dataset, such as conducting the predictions and computing the gradients. Thus, we can find that the overall speed-up is not linearly correlated with the percentage of sampled data. However, the speed-up brought by GOSS is still very significant and the technique is universally applicable to different datasets.

Second, we evaluate the accuracy of GOSS by comparing with Stochastic Gradient Boosting (SGB) [20]. Without loss of generality, we use the LETOR dataset for the test. We tune the sampling ratio by choosing different $a$ and $b$ in GOSS, and use the same overall sampling ratio for SGB. We run these settings until convergence by using early stopping. The results are shown in Table 4. We can see the accuracy of GOSS is always better than SGB when using the same sampling ratio. These results are consistent with our discussions in Sec. 3.2. All the experiments demonstrate that GOSS is a more effective sampling method than stochastic sampling.

### 5.3 Analysis on EFB

We check the contribution of EFB to the speed-up by comparing lgb_baseline with EFB_only. The results are shown in Table 2. Here we do not allow the confliction in the bundle finding process (i.e., $\gamma = 0$).[7] We find that EFB can help achieve significant speed-up on those large-scale datasets.

Please note lgb_baseline has been optimized for the sparse features, and EFB can still speed up the training by a large factor. It is because EFB merges many sparse features (both the one-hot coding features and implicitly exclusive features) into much fewer features. The basic sparse feature optimization is included in the bundling process. However, the EFB does not have the additional cost on maintaining nonzero data table for each feature in the tree learning process. What is more, since many previously isolated features are bundled together, it can increase spatial locality and improve cache hit rate significantly. Therefore, the overall improvement on efficiency is dramatic. With above analysis, EFB is a very effective algorithm to leverage sparse property in the histogram-based algorithm, and it can bring a significant speed-up for GBDT training process.

## 6 Conclusion

In this paper, we have proposed a novel GBDT algorithm called LightGBM, which contains two novel techniques: *Gradient-based One-Side Sampling* and *Exclusive Feature Bundling* to deal with large number of data instances and large number of features respectively. We have performed both theoretical analysis and experimental studies on these two techniques. The experimental results are consistent with the theory and show that with the help of GOSS and EFB, LightGBM can significantly outperform XGBoost and SGB in terms of computational speed and memory consumption. For the future work, we will study the optimal selection of $a$ and $b$ in Gradient-based One-Side Sampling and continue improving the performance of Exclusive Feature Bundling to deal with large number of features no matter they are sparse or not.

## Footnotes

[1] When we say larger or smaller gradients in this paper, we refer to their absolute values.

[2] The code is available at GitHub: `https://github.com/Microsoft/LightGBM`.

[3] Due to space restriction, high level pseudo code is used. The details could be found in our open-source code.

[4] There are some other works speed up GBDT training via GPU [17, 18], or parallel training [19]. However, they are out of the scope of this paper.

[5]Our following analysis holds for arbitrary node. For simplicity and without confusion, we omit the sub-index $O$ in all the notations.

[6]Due to space restrictions, we leave the details of parameter settings to the supplementary material.

[7]We put our detailed study on $\gamma$ tuning in the supplementary materials.

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
