[Supplementary Material · supplementary-nips.pdf]

# Supplementary materials

**Guolin Ke**[1], **Qi Meng**[2], **Thomas Finley**[3], **Taifeng Wang**[1],
**Wei Chen**[1], **Weidong Ma**[1], **Qiwei Ye**[1], **Tie-Yan Liu**[1]
[1]Microsoft Research    [2]Peking University    [3] Microsoft Redmond
[1]{guolin.ke, taifengw, wche, weima, qiwye, tie-yan.liu}@microsoft.com;
[2]qimeng13@pku.edu.cn;    [3]tfinely@microsoft.com;

This is the supplementary materials for paper "LightGBM: A Highly Efficient Gradient Boosting Decision Tree", which concludes the proofs of Theorem 3.2, proposition 2.1 and more details in the experiment section.

## 1   Theorem 3.2 and its proof

**Theorem 3.2** *Let $\bar{g}_l^j(d) = \frac{\sum_{x_i \in (A \cup A^c)_l} |g_i|}{n_l^j(d)}$ and $\bar{g}_r^j(d) = \frac{\sum_{x_i \in (A \cup A^c)_r} |g_i|}{n_r^j(d)}$. With probability at least $1 - \delta$, we have*

$$\mathcal{E}_A(d) \le C_{a,b}^2 \ln 1/\delta \cdot \max \left\{ \frac{1}{n_l^j(d)}, \frac{1}{n_r^j(d)} \right\} + 2DC_{a,b} \sqrt{\frac{\ln 1/\delta}{n}}, \tag{1}$$

*where $C_{a,b} = \frac{1-a}{\sqrt{b}} \max_{x_i \in A^c} |g_i|$, and $D = \max(\bar{g}_l^j(d), \bar{g}_r^j(d))$.*

*Proof:* For a fixed $d$, we have

$$
\begin{aligned}
&\tilde{V}_j(d) - V_j(d) \\
=&\left( \frac{(\sum_{x_i \in A_l} g_i + \frac{1-a}{b} \sum_{x_i \in B_l} g_i)^2}{n_l^j(d)} + \frac{(\sum_{x_i \in A_r} g_i + \frac{1-a}{b} \sum_{x_i \in B_r} g_i)^2}{n_r^j(d)} \right) \\
&- \left( \frac{(\sum_{x_i \in A_l} g_i + \sum_{x_i \in A_l^c} g_i)^2}{n_l^j(d)} + \frac{(\sum_{x_i \in A_r} g_i + \sum_{x_i \in A_r^c} g_i)^2}{n_r^j(d)} \right) \\
=&C_l \left( \frac{1-a}{b} \sum_{x_i \in B_l} g_i - \sum_{x_i \in A_l^c} g_i \right) + C_r \left( \frac{1-a}{b} \sum_{x_i \in B_r} g_i - \sum_{x_i \in A_r^c} g_i \right)
\end{aligned}
$$

where $C_l = \frac{\left( \frac{1-a}{b} \sum_{x_i \in B_l} g_i + \sum_{x_i \in A_l^c} g_i + 2(\sum_{x_i \in A_l} g_i) \right)}{n_l^j(d)}$,

and $C_r = \frac{\left( \frac{1-a}{b} \sum_{x_i \in B_r} g_i + \sum_{x_i \in A_r^c} g_i + 2(\sum_{x_i \in A_r} g_i) \right)}{n_r^j(d)}$.

Thus, we have

$$|\tilde{V}_j(d) - V_j(d)| \le \max\{C_l, C_r\} \left| \frac{1-a}{b} \sum_{x_i \in B} g_i - \sum_{x_i \in A_c} g_i \right| \tag{2}$$

Firstly, we bound $C_l$ and $C_r$. Let $D_{A^c} = \max_{x_i \in A^c} |g_i|$, we have

$$C_l = \frac{\left(\frac{1-a}{b}\sum_{x_i \in B_l} g_i + \sum_{x_i \in A_l} g_i\right)}{n_l^j(d)} + \frac{\left(\sum_{x_i \in A_l^c} g_i + \sum_{x_i \in A_l} g_i\right)}{n_l^j(d)} \tag{3}$$

$$\leq \frac{D_{A^c}\left|\frac{1-a}{b}\sum I_{[x_i \in B_l]} - \sum I_{[x_i \in A_l^c]}\right|}{n_l^j(d)} + 2D \tag{4}$$

$$= \frac{D_{A^c}(1-a)n}{n_l^j(d)}\left|\frac{\sum I_{[x_i \in B_l]}}{bn} - \frac{\sum I_{[x_i \in A_l^c]}}{(1-a)n}\right| + 2D \tag{5}$$

By Hoeffding's inequality, we have with probability at least $1 - \delta$,

$$C_l \leq \frac{D_{A^c}(1-a)n}{n_l^j(d)}\sqrt{\frac{\ln 2/\delta}{2bn}} + 2D. \tag{6}$$

Similarly, we have $C_r \leq \frac{D_{A^c}(1-a)n}{n_r^j(d)}\sqrt{\frac{\ln 2/\delta}{2bn}} + 2D$.

For the term $\left(\frac{1-a}{b}\sum_{x_i \in B} g_i - \sum_{x_i \in A^c} g_i\right)$, we have with probability at least $1 - \delta$,

$$\frac{1}{n}\left|\frac{1-a}{b}\sum_{x_i \in B} g_i - \sum_{x_i \in A^c} g_i\right| \leq D_{A^c}(1-a)\sqrt{\frac{\ln 2/\delta}{2bn}}. \tag{7}$$

Thus, we have with probability at least $1 - \delta$

$$\mathcal{E}(d) = \left|\frac{\tilde{V}_j(d)}{n} - \frac{V_j(d)}{n}\right|$$

$$\leq \left(D_{A^c}(1-a)\max\left\{\frac{1}{n_l^j(d)}, \frac{1}{n_r^j(d)}\right\}\sqrt{\frac{n\ln 1/\delta}{2b}} + 2D\right)a)\sqrt{\frac{\ln 1/\delta}{2bn}}$$

$$\leq \frac{D_{A^c}^2(1-a)^2\ln 1/\delta}{2b}\cdot\max\left\{\frac{1}{n_l^j(d)}, \frac{1}{n_r^j(d)}\right\} + \frac{2D\cdot D_{A^c}\cdot(1-a)}{\sqrt{2b}}\sqrt{\frac{\ln 1/\delta}{n}}.$$

Putting $C_{a,b}$ in the above inequality, we can get the result in the theorem. $\square$

**Discussions:**

(1) The high probability error given in the above theorem is related to $n, n_l^j(d), n_r^j(d), C_{a,b}$. The asymptotic rate of the approximated gain to the original gain is $\mathcal{O}\left(\frac{1}{n_l^j(d)} + \frac{1}{n_r^j(d)} + \frac{1}{\sqrt{n}}\right)$. If the number of instances in the two subsets after splitting are relatively balanced (i.e., $n_l^j(d) \geq \mathcal{O}(\sqrt{n})$ and $n_r^j(d) \geq \mathcal{O}(\sqrt{n})$), the approximation error will be dominated by the second term of Ineq.(1) and it will decrease as $n$ becomes large.

(2) For fixed $b$, as $a$ becomes larger, $C_{a,b}$ will become smaller because both term $1 - a$ and term $\max_{x_i \in A^c} |g_i|$ will become smaller. For fixed $a$, as $b$ becomes larger, $\frac{1}{\sqrt{b}}$ will become smaller and the upper bound will become smaller.

(3) Random sampling is a special case for GOSS with $a = 0$. In many cases, GOSS could outperform random sampling. More specifically, the condition is $C_{0,\beta} > C_{a,\beta-a}$, which is equivalent to $\frac{\alpha_a}{\sqrt{\beta}} > \frac{1-a}{\sqrt{\beta-a}}$ with $\alpha_a = \max_{x_i \in A \cup A^c} |g_i| / \max_{x_i \in A^c} |g_i|$. Thus the condition for GOSS to achieve better accuracy is $\beta > \frac{1}{1-((1-a)/\alpha_a)^2}\cdot a$. For fixed $\beta$ which satisfies the condition, if $\alpha_a$ is increasing rapidly as $a$ is increasing, GOSS prefers a larger $a$. If $\alpha_a$ is increasing smoothly as $a$ is increasing, GOSS prefers smaller $a$. For example, if $a_0, a_1, a_2$ are the smallest $a$ to make $\alpha_{a_0} \geq \sqrt{2}, \alpha_{a_1} \geq \sqrt{3}, \alpha_{a_2} \geq 2$, then the condition for $\beta$ is $\beta > 2a_0, \beta > 1.5a_1, \beta > 1.25a_2$, respectively. $\alpha_a$ is increasing rapidly means that $a_2 - a_1$ and $a_1 - a_0$ are small values, which makes $1.25a_2$ smaller than $2a_0$ and $1.5a_1$. In this case, GOSS prefers $a_2$.

(4) Theorem 3.2 is established for any fixed split point. Thus we can bound $|\tilde{V}(d^*) - V(d^*)|$. It is easier to bound $|\tilde{V}_j(\tilde{d}^*) - \tilde{V}_j(d^*)|$ by using the techniques in [1, 2]. Combining them we can bound $|\tilde{V}_j(\tilde{d}^*) - V(d^*)|$, which is the difference between the largest variance gain calculated by the original set $A \cup A^c$ and that calculated by the set $A \cup B$.

## 2   Accuracy guarantee for random polluting

We denote the maximal variance gain for feature $j$ as $V_j = \max_d V_j(d)$ and the maximal variance gain as $V = \max_j V_j$. We denote the maximal variance gain with random polluting with polluting rate $\gamma$ as $V^\gamma$. Assume that the maximal variance gain is achieved at feature $j_1$, i.e., $V = V_{j_1}$ and the maximal variance gain with random polluting is achieved at feature $j_2$, i.e., $V^\gamma = V_{j_2}^\gamma$, we have the following theorem.

**Proposition 2.1** *The difference of the maximal variance gain calculated between with and without random polluting can be bounded as below:*

$$|V - V^\gamma| \le [(1-\gamma)n]^{-\frac{2}{3}}, \tag{8}$$

*where $n$ is the number of training instances.*

*Proof:* Since $V = V_{j_1} \ge V_{j_2}$, and $V^\gamma = V_{j_2}^\gamma \ge V_{j_1}^\gamma$, we have

$$V_{j_1} - V_{j_2}^\gamma \le V_{j_1} - V_{j_1}^\gamma \tag{9}$$

$$V_{j_2}^\gamma - V_{j_1} \le V_{j_2}^\gamma - V_{j_2}. \tag{10}$$

Using the results in [1], we have $V_{j_1} - V_{j_1}^\gamma \le [(1-\gamma)n]^{-\frac{2}{3}}$ and $V_{j_2}^\gamma - V_{j_2} \le [(1-\gamma)n]^{-\frac{2}{3}}$. Thus we can get the results in the theorem.

## 3   More details in experiments

### 3.1   Detailed parameter settings in experiments

The details of parameter settings used in experiments are listed in Table 1, Table 2 and Table 3. As we use XGBoost as baseline, we also use the same parameter name in their documents. For meanings of these parameters, please refer to `https://github.com/dmlc/xgboost/blob/master/doc/parameter.md`. For the SGB experiments, as less data are used in training, we need to reduce the min_child_weight and min_child_data accordingly to avoid under-fitting.

The settings of $a$ and $b$ on the Table 4 of Experiments section in the main paper are listed on Table 4. We tune the combination of $a$, $b$ under the overall sampling ratio constraint, and the listed settings can produce relatively good performance on the model accuracy.

Table 1: Common Settings.

| | |
|---|---|
| Allstate | learning_rate=0.02, min_child_weight=100, num_round=500 |
| Flight Delay | learning_rate=0.1, min_child_weight=100, num_round=1000 |
| LETOR | learning_rate=0.05, min_child_weight=100, num_round=1000 |
| KDD10 | learning_rate=0.1, min_child_weight=3000, num_round=100 |
| KDD12 | learning_rate=0.1, min_child_weight=3000, num_round=100 |

Table 2: Settings for xgb_exa.

| | |
|---|---|
| Allstate | max_depth=12, min_split_gain=0.5 |
| Flight Delay | max_depth=12, min_split_gain=60 |
| LETOR | max_depth=16 |
| KDD10 | max_depth=50, min_split_gain=150 |
| KDD12 | max_depth=37, min_split_gain=100 |

Table 3: Settings for xgb_his, lgb_baseline, SGB and LightGBM.

| | |
|---|---|
| Allstate | num_leaves=127 |
| Flight Delay | num_leaves=255 |
| LETOR | num_leaves=255 |
| KDD10 | num_leaves=255, min_child_data=1000 |
| KDD12 | num_leaves=255, min_child_data=1000 |

Table 4: Settings of GOSS in Table 4 of Experiments section in the main paper.

| Sampling ratio | 0.1 | 0.15 | 0.2 | 0.25 | 0.3 | 0.35 | 0.4 |
|---|---|---|---|---|---|---|---|
| GOSS | a=0.05 b=0.05 | a=0.05 b=0.1 | a=0.1 b=0.1 | a=0.15 b=0.1 | a=0.2 b=0.1 | a=0.25 b=0.1 | a=0.15 b=0.25 |

## 3.2 Effect of $\gamma$ in EFB

We evaluate the influence of using different $\gamma$ in EFB on KDD10 and KDD12 datasets. The results are shown in Table 5 and Table 6. The accuracy drops if using a relative big $\gamma$, e.g. using $\gamma = 0.01$ causes the accuracy on KDD2010 dropping from 0.7873 to 0.7858. And the accuracy is almost the same as baseline if using small $\gamma$. This is consistent with our theoretical analysis. We also notice the speed-up brought by enabling bundling confliction is small. It is because EFB already bundle many sparse features into very few dense features, and allowing conflict cannot further reduce it significantly. So, in our main paper, we just set $\gamma = 0$. However, this does not affect the effectiveness of EFB. If the data contains many slightly conflicted sparse features, our algorithm with small $\gamma$ will be significantly faster.

Table 5: Effect on accuracy by using different $\gamma$ in EFB.

|  | baseline | $\gamma = 0.01$ | $\gamma = 0.001$ | $\gamma = 0.0001$ | $\gamma = 0.00001$ |
|---|---|---|---|---|---|
| KDD10 | 0.78735 | 0.785806 | 0.787218 | 0.78738 | 0.787391 |
| KDD12 | 0.704854 | 0.703197 | 0.704606 | 0.704612 | 0.704757 |

Table 6: Effect on speed by using different $\gamma$ in EFB.

|  | baseline | $\gamma = 0.01$ | $\gamma = 0.001$ | $\gamma = 0.0001$ | $\gamma = 0.00001$ |
|---|---|---|---|---|---|
| KDD10 | 6.33 | 5.88 | 6.12 | 6.25 | 6.29 |
| KDD12 | 20.23 | 19.06 | 19.78 | 20.04 | 20.18 |

## 3.3 Time-accuracy curves for all datasets

In order to conduct end-to-end comparison between all algorithms, we draw the time-accuracy curves on all experiment datasets. The curves are shown in Fig. 1, Fig.2, Fig.3, Fig.4 and Fig.5.

LightGBM demonstrates good performance on all datasets. It is much faster than the other baselines while achieving the best accuracy. As KDD10 and KDD12 datasets are too large, other baselines cannot converge in reasonable time. So, our accuracy results for these two datasets on the Table 3 in the main paper are the accuracy of 100-th iteration. However, since LightGBM is much faster than other tools, we did not limit the number of iterations in Fig.4 and Fig.5. So, its convergence points are better than the values of Table 3 in the main paper.

Figure 1: Time-accuracy on Allstate dataset.    Figure 2: Time-accuracy on Flight Delay dataset.

Figure 3: Time-accuracy on LETOR dataset.

Figure 4: Time-accuracy on KDD10 dataset.

Figure 5: Time-accuracy on KDD12 dataset.