[Reviews · NeurIPS 2017]

Reviewer 1



The paper presents two nice ways for improving the usual gradient boosting algorithm where weak classifiers are decision trees. It is a paper oriented towards efficient (less costful) implementation of the usual algorithm in order to speed up the learning of decision trees by taking into account previous computations and sparse data. The approaches are interesting and smart. A risk bound is given for one of the improvements (GOSS), which seems sound but still quite loose: according to the experiments, a tighter bound could be obtained, getting rid of the "max" sizes of considered sets. No garantee is given for the second improvement (EFB) although is seems to be quite efficient in practice. The paper comes with a bunch of interesting experiments, which give good insights and shows a nice study of the pros and vons of the improving approaches, both in accuracy and computational time, in various types of datasets. The experiments lacks of standard deviation indications, for the performances are often very closed from one mathod to another. A suggestion would be to observe the effect of noise in the data on the improvement methods (especially on GOSS, which relies mostly on a greedy selection based on the highest gradients). The paper is not easy to read, because the presentation, explanation, experimental study and analysis are spread all over the paper. It would be better to concentrate fully on one improvements, then on the other, and to finish by the study of them together. Besides, it would be helpfulfor the reader if the main algorithms where written in simplest ways, not in pseudo codes and assumption on the specificities of data structures.

Reviewer 2



This paper investigates the gradient boosting decision tree (GBDT) in machine learning. This work focuses on the case when the feature dimension is high and data size is large, and previous studies are not unsatisfactory with respect to efficiency and scalability. In this paper, the authors introduce two techniques: Gradient-based One-Side Sampling (GOSS) and Exclusive Feature Bundling (EFB). GOSS excludes a significant proportion of data instances with small gradients, and only uses the rest to estimate the information gain. EFB bundles mutually exclusive features (i.e., they rarely take nonzero values simultaneously), to reduce the number of features. The authors prove that GOSS can obtain quite accurate estimation of the information gain with a much smaller data size, and finding the optimal bundling of exclusive features is NP-hard, but a greedy algorithm can achieve quite good approximation ratio. Finally, the some experiments are presented to show that LightGBM speeds up the training process of conventional GBDT while keeping almost the same accuracy. Good points: 1. This paper is well-motivated, and the proposed algorithm is interesting and effective. 2. The authors present the theoretical analysis for approximation error. 3. The authors provide strong empirical studies to shows the effectiveness of the proposed algorithm, which present good insights in various datasets. It would be better to provide some details on the datasets and why these datasets are selected. It would be better to provide the generalization analysis since the algorithm is measured by test accuracy and AUC.

Reviewer 3



This paper gives LightGBM, an improvement over GBDT which is of greate value in practical applicaitons. Two techniques (i.e., GOSS and EFB) are proposed to tackle the computational problem of estimating information gain on large datasets. Specifically, at each step, GOSS excludes a large proportion of instances with small gradients; and EFB bundles mutually exclusive features to reduce the number of features. Theoretical analysis and empirical studies show the effectiveness of the proposed methods. Strong points: 1. Overall, the paper is well written and easy to follow. 2. The proposed studied here is well motivated and of important pratical value, and the proposed methods are smart and effective. 3. It is nice to see theoretical analysis for the GOSS method. 4. The experiment results are convincing, eitheri n accuracy or computational time. It is pity to see that the theoretical study is on the approximation error, it would be better to see generalization error analysis. That is, how the approximation error will be affect the generalization error.